# Simulation of an offshore wind farm using fluid power for centralized electricity generation

Antonio Jarquin Laguna

Section Offshore Engineering, Faculty of Civil Engineering and Geosciences, Delft University of Technology, Stevinweg 1, 2628 CN, Delft, The Netherlands

*Correspondence to:* A.JarquinLaguna@tudelft.nl

**Abstract.** A centralized approach for electricity generation within a wind farm is explored through the use of fluid power technology. This concept considers a new way of generation, collection and transmission of wind energy inside a wind farm, in which electrical conversion does not occur during any intermediate conversion step before the energy has reached the offshore central platform. A numerical model was developed to capture the relevant physics from the dynamic interaction between

different turbines coupled to a common hydraulic network and controller. This paper presents a few examples of the time-domain simulation results for a hypothetical hydraulic wind farm subject to turbulent wind conditions. The performance and operational parameters of individual turbines are compared with those of a reference wind farm based on conventional wind turbine generator technology using the same wind farm layout and environmental conditions. For the presented case studies, results indicate that the individual wind turbines are able to operate within operational limits. Despite the stochastic turbulent

wind conditions and wake effects, the hydraulic wind farm is able to produce electricity with reasonable performance in both below and above rated conditions. With the current pressure control concept, a continuous operation of the hydraulic wind farm is shown including the full stop of one or more turbines.

## 1 Introduction

A typical offshore wind farm consists of an array of individual wind turbines several kilometers from shore. Each of these

turbines captures the kinetic energy from the wind and converts it into electrical power in a similar way as is done with onshore technology. However, one main characteristic of a wind farm as a collection of individual turbines, is that electricity is still generated in a distributed manner. This means that the whole process of electricity generation occurs separately and the electricity is then collected, conditioned and transmitted to shore. When looking at a wind farm as a power plant, it seems reasonable to consider the use of only a few generators of larger capacity rather than around one hundred of generators of lower

capacity. The potential benefits, challenges and limitations of a centralized electricity generation scheme for an offshore wind farm are not known yet.

This work explores a particular concept in which a centralized electricity generation within a wind farm is proposed by means of a hydraulic network using fluid power technology (Diepeveen, 2013). The basic idea behind the concept is to dedicate the individual wind turbines to create a pressurized flow of seawater. Then, the flow is collected from the turbines and redirected

through a network of pipelines to a central generator platform. At the platform, the overall pressurized flow is converted first into mechanical and later into electrical power through an impulse hydraulic turbine. Modern hydro-turbines have been developed with typical capacities of 500 MW operating for decades with enough operational and maintenance experience gained from conventional hydro-power plants. On the other hand using hydro turbines in combination with renewable energy sources such as offshore wind energy has not been explored in-depth so far.

The main motivation for introduction of a centralized offshore wind farm is to reduce the complexity and capital cost for the individual rotor nacelle assemblies. It is also expected that by having the whole electrical generation equipment in one offshore central platform instead of having it in a constraint space hundred meters above sea level, would have a positive impact regarding operation and maintenance costs. A conceptual comparison between a conventional and the proposed offshore wind farm is shown in Fig. 1.

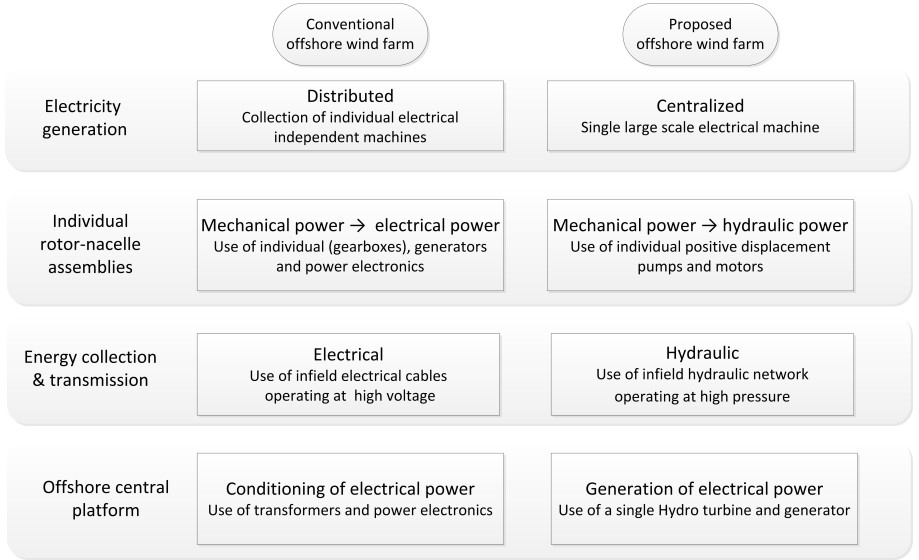

**Figure 1.** Conceptual comparison between a conventional and the proposed offshore wind farm.

Hydraulic systems have already shown their effectiveness when used for demanding applications where performance, durability and reliability are critical aspects. In particular, the efficient and easy generation of linear movements, together with their good dynamic performance give hydraulic drives a clear advantage over mechanical or electrical solutions. Furthermore, hydraulic drives have the potential to facilitate the integration with energy storage devices such as hydraulic accumulators which are important to smooth the energy output from wind energy applications (Innes-Wimsatt and Loth, 2014). In any industry where robust machinery is required to handle large torques, hydraulic drive systems are a common choice. They have a long and successful track record of service in, for example, mobile, industrial, aircraft and offshore applications (Cundiff, 2001; Albers, 2010). Therefore, it is evident that the use of hydraulic technology is recognized as an attractive alternative solution for power conversion in wind turbines (Salter, 1984).

For the proposed concept, using high pressure makes it possible to reduce the top mass of the individual rotor-nacelle assemblies. For this reason, a high potential exists to reduce the amount of structural steel needed in the support structures as well; for a 5 MW turbine in 30 m water depth, 1.9 ton of structural steel of the monopile can be saved for every ton of top mass reduction (Segeren and Diepeveen, 2014). Using high pressures makes the use of fluid power an attractive means to transmit the captured energy from the rotor-nacelle assemblies to a central platform.

With the purpose to avoid fluid circulation, an open-loop circuit is considered with seawater as hydraulic fluid. The choice of seawater as hydraulic fluid is preferred because of its availability and environmental friendly nature when compared to oil hydraulics. In this regard, it is important to consider that seawater contains a high concentration of minerals, which give it a high degree of hardness. It also contains dissolved gases such as oxygen and chlorine which cause corrosion. Despite its corrosive nature, the use of seawater hydraulics has already been used in some industrial applications, where in terms of safety, water hydraulics might be preferred due to potential fire hazards or risk of leakage as is the case of the mining industry. An example in the offshore industry includes the seawater hydraulic system for deep sea pile driving incorporating high pressure water pumps (Schaap, 2012). A key advantage of this system is that the use of an open-loop circuit cancels the need for cooling equipment, a disadvantage is that it is likely that filters have to be cleaned more frequently.

The modelling and analysis of a single turbine with hydraulic technology has been previously presented for variable-speed control strategies. Simulations of an individual turbine with an oil based hydrostatic transmission have been presented in (Jarquin Laguna et al., 2014). The results showed good dynamic behaviour for turbulent wind conditions where reduced fluctuations of the drivetrain torque and power are obtained despite the reduced energy capture. The integration of a single turbine with a Pelton runner using water hydraulics was introduced in (Jarquin Laguna, 2015), where a passive variable speed strategy was proposed. However, the addition and simulation of more turbines to the hydraulic network was not included. In an effort to assess the trade-offs implied by the proposed hydraulic concept, this paper extends the time-domain simulations to evaluate the performance and operational parameters of five turbines coupled to a common hydraulic network for a hypothetical wind farm with centralized electricity generation. In the first part of this work, an overview of the wind farm model is presented together with the control strategy of the hydraulic components; the second part describes a case example where the results are compared with those of a typical wind farm based on conventional wind turbine generator technology.

## 2   Wind farm model overview

The overall wind farm model, incorporates the dynamic interaction between the individual turbines, the hydraulic network, the Pelton turbine and the controller. The model is described as a set of coupled algebraic and non-linear ordinary differential equations which are solved by numeric integration using Matlab-Simulink. The hydraulic wind power plant model is composed by the following subsystems:

## 2.1  Wind turbines

### 2.1.1  Aerodynamic model

The aerodynamic characteristics of a horizontal axis wind turbine rotor are a function of its rotational speed $\omega_r$, the pitch angle of the blades $\beta$ and the relative velocity of the upstream wind speed $U$ with respect to the rotor. The aerodynamic torque $\tau_{aero}$, axial thrust $F_{thrust}$ and power $P_{aero}$ are described through their non-dimensional steady-state performance coefficients as a function of the upstream wind speed.

$$\tau_{aero} = C_\tau(\lambda,\beta)\, \frac{1}{2}\, \rho_{air}\, \pi\, R^3\, U_{rel}^2 \tag{1}$$

$$F_{thrust} = C_{Fax}(\lambda,\beta)\, \frac{1}{2}\, \rho_{air}\, \pi R^2\, U_{rel}^2 \tag{2}$$

$$P_{aero} = C_P(\lambda,\beta)\, \frac{1}{2}\, \rho_{air}\, \pi R^2\, U_{rel}^3 \tag{3}$$

where $\rho_{air}$ is the air density, $R$ is rotor radius and the tip speed ratio $\lambda$ is defined as the ratio of the tangential velocity of the blade tip and the upstream undisturbed wind speed.

$$\lambda = \frac{\omega_r\, R}{U_{rel}} \tag{4}$$

This reduced order model does not include any aero-elastic or unsteady aerodynamic effects. Although these aspects are important for the loading of both rotor and support structure, their effects on the aerodynamic torque are considered less relevant from the performance and control point of view of the overall wind farm. The relatively large mass moment of inertia of the rotor in the angular degree of freedom will absorb large peak fluctuations in the rotor speed derived from the unsteady aerodynamic effects on the rotor torque.

### 2.1.2  Hydraulic drive train model

The hydraulic drive train consists of a large positive displacement water pump directly coupled to the low-speed rotor shaft. Hence, the rotor-pump angular acceleration is described through the balance of the aerodynamic torque $\tau_{aero}$, and the transmitted torque from the pump $\tau_p$ as a first order differential equation. The mass moment of inertia of the rotor and pump is described by:

$$J_r\, \dot{\omega}_r - \tau_{aero}(U,\beta,\omega_r) + \tau_p(\omega_r,\Delta p_p, V_p) = 0 \tag{5}$$

The pump is mainly characterized through a variable volumetric displacement $V_p$, which determines the volume of fluid that is obtained for each rotor revolution. Hence the volumetric flow rate of the pump $Q_p$ is ideally given by the product of its volumetric displacement and the rotor shaft speed; internal leakage losses are included as a linear function of the pressure drop

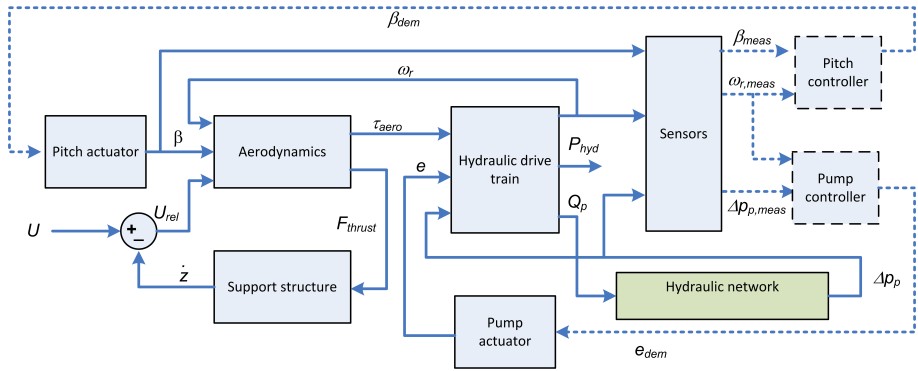

**Figure 2.** Subsystem block diagram of a single turbine connected to the hydraulic network.

across the pump $\Delta p$ with the laminar leakage coefficient $C_s$. In a similar manner, the transmitted torque is directly related to the volumetric displacement and the pressure across the pump; a friction torque is described with a viscous and a dry component defined with the damping coefficient $B_p$ and a friction coefficient $C_f$ respectively (Merritt, 1967).

$$Q_p = V_p\,\omega_r - C_s\,\Delta p_p \tag{6}$$

$\quad \tau_p = V_p\,\Delta p_p + B_p\,\omega_r + C_f\,V_p\,\Delta p_p \tag{7}$

Here $e$ is introduced as the ratio of the current volumetric displacement and its nominal value per rotational cycle such that:

$$V_p\,(e) = e\,V_{p,max} \tag{8}$$

The variable $e$ from Eq. (8) is used as a control variable to modify either the volumetric flow rate or the transmitted torque of the pump. The dynamics of a general actuator used to modify the volumetric displacement of the pump are approximated

by a first order differential equation. The constant $T_e$ characterizes how slow or fast the actuator responds to a reference value input $e_{dem}$ according to the following equation:

$$\dot{e} = \frac{1}{T_e}\,(e_{dem} - e) \tag{9}$$

The yaw degree of freedom of the individual turbines is not considered. Hence, the yaw controller of the turbines is not included. A schematic showing the different subsystems of a single turbine is shown in Fig. 2.

**2.1.3 Pitch actuator model**

The pitch actuator is based on a pitch-servo model described by a proportional regulator with constant $K_\beta$. The demanded pitch $\beta_{dem}$ is obtained from the signal of the pitch controller. The second order model includes a time constant $t_\beta$ and an input

delay from input $u_\beta$ to the pitch rate $\dot{\beta}$. During the simulation, the delayed input $u_\beta^\delta$ is implemented by storing the input signal and the simulation time in a buffer for a specified amount of time given by $\delta$. The pitch actuator is implemented with pitch rate limits of $\pm 8\,^\circ$.

$$\ddot{\beta} = \frac{1}{t_\beta} \left( u_\beta^\delta - \dot{\beta} \right) \tag{10}$$

$u_\beta = K_\beta \left( \beta_{dem} - \beta_{meas} \right)$ (11)

### 2.1.4   Structural model

The motion of the top mass of the tower in the fore-aft direction $z$ is described with a second order model:

$$m_{tm}\,\ddot{z} = F_{thrust} - B_{tower}\,\dot{z} - K_{tower}\,z \tag{12}$$

where $K_{tower}$ and $B_{tower}$ are the support structure stiffness and damping; $F_{thrust}$ is the thrust force exerted by the rotor on
the top mass of the tower $m_{tm}$, which includes the rotor and nacelle mass. The thrust force is calculated through Eq. (2) using the tip speed ratio from Eq. (4) and the rotor speed obtained from the solution of Eq. (5).

### 2.2   Hydraulic network

One of the key aspects for having a centralized electricity generation is the use of hydraulic networks to collect and transport the pressurized water from the individual wind turbines to the generator platform. Similarly to the electrical inter-array cable system
for a conventional offshore wind farm, the design of the hydraulic lay-out should consider several practical and economical aspects, such as reducing the number and length of pipelines, operational losses and installation methods. For wind farms with a large number of turbines, it is expected that branched hydraulic networks using parallel and common pipelines will result in the most convenient configuration. The hydraulic network consists of a number of interconnected pipelines represented by linear transmission line models. The approach to construct this network for time-domain simulations from individual pipelines was
previously presented in (Jarquin Laguna, 2014). The linear models are only given for laminar flow, for steady flow the criteria for occurrence of turbulence is simply given by the Reynolds number; however, for unsteady flow neither the criteria used to predict flow instability, nor the manner in which it occurs is well understood. In the case of an oscillating flow component which is superimposed on a mean turbulent flow, the laminar flow solutions might be still applicable over a limited turbulent flow range. Both physical and empirical-based corrections to the shear stress model have been proposed for turbulent pipe
transients (Vardy et al., 1993; Vardy and Brown, 1995). The correct modelling of turbulence in transient flows is an ongoing research topic; it is not addressed in this work.

The dynamic response of the compressible laminar flow of a Newtonian fluid through a rigid pipeline network is given by the following state-space model; the model includes inertia and compressibility effects which are necessary to describe the

fluid transients or so-called 'water-hammer' effects. The model uses the volumetric flow rates from the individual rotor driven pumps and at the nozzle as an input, and the pressures at the water pumps and nozzle as an output.

$$\text{Hydraulic network model} \left\{ \quad \dot{\mathbf{x}} = \mathbf{A_Q}\mathbf{x} + \mathbf{B_Q} \begin{pmatrix} Q_{p,1} \\ Q_{p,2} \\ \vdots \\ Q_{p,i} \\ Q_{nz} \end{pmatrix}, \quad \begin{pmatrix} \Delta p_{p,1} \\ \Delta p_{p,2} \\ \vdots \\ \Delta p_{p,i} \\ \Delta p_{nz} \end{pmatrix} = \mathbf{C_Q}\mathbf{x} \right. \tag{13}$$

The matrices $\mathbf{A_Q}$, $\mathbf{B_Q}$ and $\mathbf{C_Q}$ are defined in terms of the physical parameters of the hydraulic lines and water properties such as water viscosity, water density, speed of sound in the water, length and internal radius of the pipelines. A schematic of the model showing the input-output causality for each element is shown in Fig. 3.

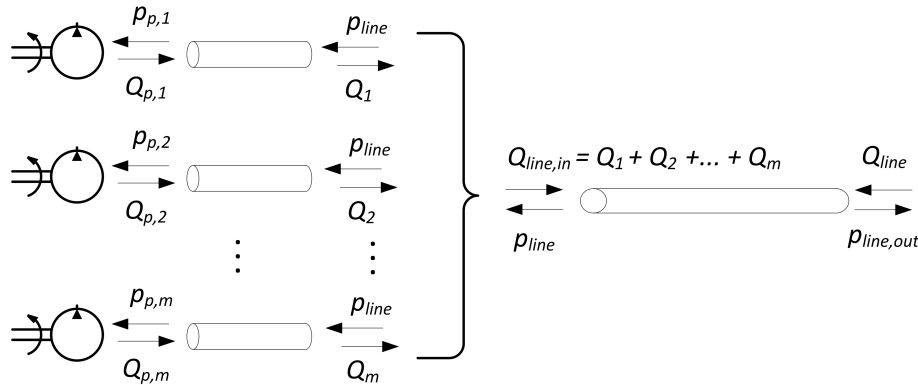

**Figure 3.** Schematic for parallel hydraulic lines connected to a common line.

### 2.3 Nozzle and spear valve

At the end of the hydraulic network, a nozzle and spear valve is used to adapt the pressurized water flow into the Pelton turbine. The nozzle characteristics are included as a first order differential equation by taking the momentum equation of a fluid particle into account along the nozzle length $L_{nz}$ as described in Eq. (14), (Makinen et al., 2010).

$$\rho_{hyd} L_{nz} \dot{Q}_{nz} = \Delta p_{nz} A_{nz}(h_s) - \frac{\rho_{hyd} Q_{nz} |Q_{nz}|}{2 A_{nz}(h_s) C_d^2} \tag{14}$$

Where $\rho_{hyd}$ is the density of the hydraulic fluid, $A_{nz}$ is the nozzle cross sectional area determined by the position of the spear valve, and $C_d$ is the discharge coefficient to account for pressure losses due to the geometry and flow regime at the nozzle exit. The nozzle cross sectional area is described by the linear position of the spear valve $h_s$ according to Eq. (15). It is assumed

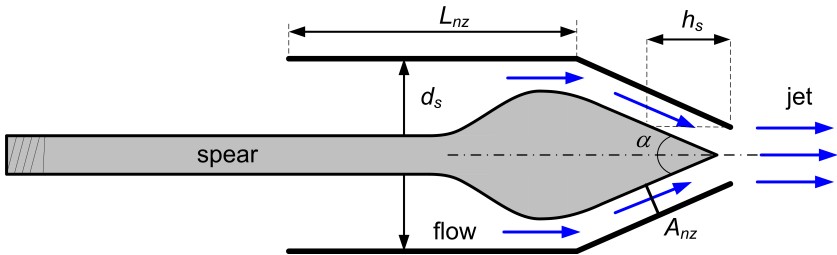

**Figure 4.** Schematic of the spear valve and nozzle.

that the spear valve position is smaller than the fixed nozzle diameter $d_s$. The geometric characteristics of the spear valve are included through the spear cone angle $\alpha$ as shown in Figure 4.

$$A_{nz}\left(h_s\right) = \min\left(\pi\left[h_s\, d_s\, \sin\left(\frac{\alpha}{2}\right) - h_s^2 \sin^2\left(\frac{\alpha}{2}\right)\cos\left(\frac{\alpha}{2}\right)\right], \frac{\pi}{4}d_s^2\right)) \tag{15}$$

Figure 5 shows the normalized cross sectional area of the nozzle as function of the spear valve linear position for different spear cone angles.

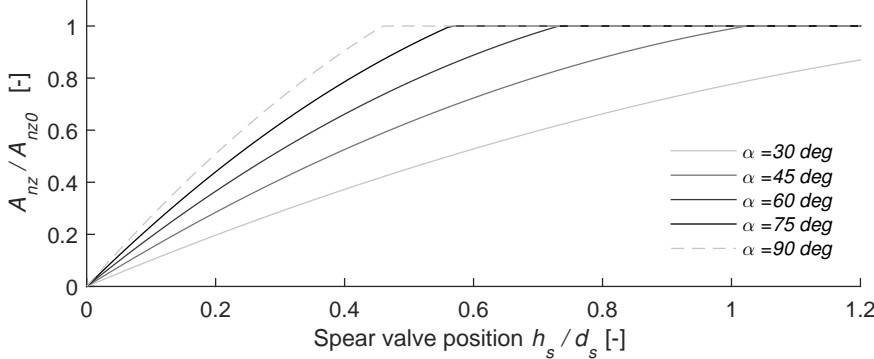

**Figure 5.** Cross sectional area of the nozzle as function of the spear valve linear position for different spear cone angles where $A_{nz0} = \frac{\pi}{4}d_s^2$.

Similarly to the pump actuator, the dynamics of the spear valve linear actuator are approximated by a first order differential equation in which a constant $T_h$ characterizes how slow or fast the spear valve position responds to reference value input $h_{s,dem}$ according to the following equation:

$$\dot{h}_s = \frac{1}{T_h}\left(h_{s,dem} - h_s\right) \tag{16}$$

The hydraulic power at the nozzle $P_{hyd}$ is given by the product of the volumetric flow rate and the water pressure at this location.

$$P_{hyd} = Q_{nz}\,\Delta p_{nz} \tag{17}$$

## 2.4 Pelton turbine

The hydraulic efficiency of the Pelton runner $\eta_P$ is obtained from momentum theory according to different geometrical and operational parameters as described in (Thake, 2000) and (Zhang, 2007).

$$\eta_P = 2k\,(1-k)\,(1-\xi\cos\gamma) \tag{18}$$

where $\xi$ is an efficiency factor to account for the friction of the flow in the bucket, $\gamma$ is defined as the angle between the circumferential and relative velocities, and $k$ is the runner speed ratio defined by the ratio between the tangential velocity of

the runner at Pitch Circle Diameter (PCD) and the water jet speed $U_{jet}$.

$$k = \frac{\omega_P R_{PCD}}{U_{jet}} \tag{19}$$

The theoretical Pelton efficiency is shown in Fig. 6 for different friction factors and constant bucket angle. Optimal efficiency is obtained when the water jet velocity is twice the tangential velocity of the runner at PCD. If the Pelton runner speed is kept constant, then the jet velocity and hence the pressure drop across the nozzle should be also kept constant in order to operate at

maximum efficiency. A Pelton turbine operating with a constant rotational speed considerably simplifies the integration with the electrical grid. The constant rotational speed is realized by using a grid-connected synchronous generator, similar to most large scale hydroelectric generation plants.

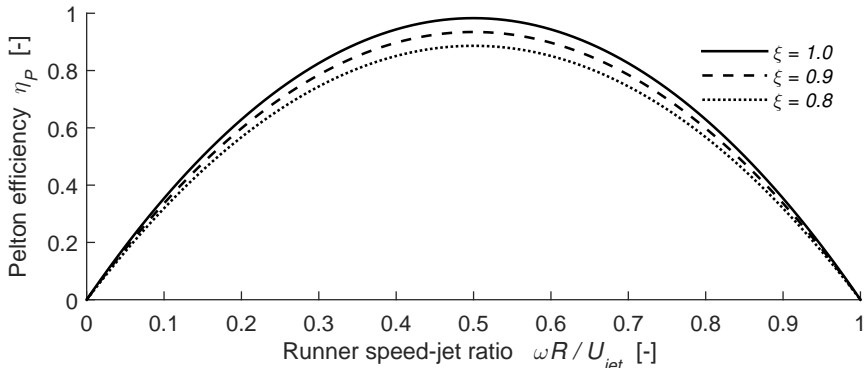

**Figure 6.** Theoretical Pelton efficiency for different values of friction factor $\xi$ and $\gamma = 165$ degrees.

For the proposed configuration the efficiency of the Pelton turbine is only determined by the water jet velocity, which is simply the volumetric flow rate divided by the cross sectional area and multiplied by a vena contracta coefficient $C_v$ to account for the change in velocity immediately after the water jet exits the nozzle. The vena contracta phenomenon does not influence the nozzle efficiency and a coefficient value of 0.99 was used according to (Thake, 2000).

$$5 \quad U_{jet} = C_v \frac{Q_{nz}}{A_{nz}(h_s)} \qquad (20)$$

## 2.5 Environmental conditions

The dynamic wind flow models and wake effects for a given layout are based on an open source toolbox developed for 'Distributed Control of Large-Scale Offshore Wind Farms' as part of the European FP7 project with the acronym Aeolus (Grunnet et al., 2010). The model assumes a 2D wind field generated at the hub height plane. The wind field does not account
10    for wind shear or tower shadow effects and is generated at hub height plane. The mean wind speed has a constant value in the longitudinal direction and zero lateral component. Similarly, the wind speed direction is fixed with respect to the farm layout in longitudinal direction. The turbulent wind field is generated using a Kaimal spectrum; two spectral matrices together with coherence parameters are used to describe the spatial variations of the wind speed according to (Veers, 1988).

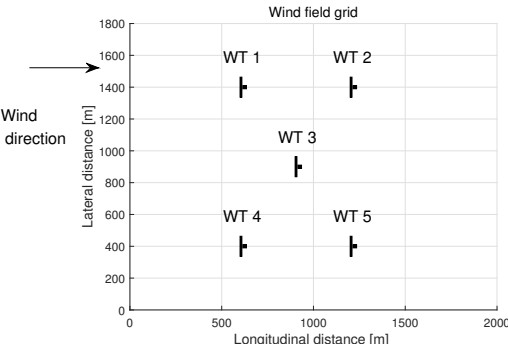

**Figure 7.** Layout of the proposed wind farm with five turbines of 5MW each.

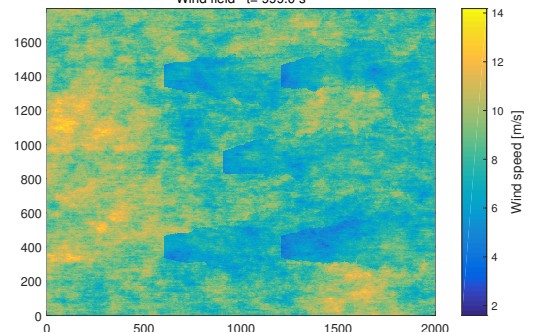

**Figure 8.** Snap shot of the wind field and wake effects.

Three wake effects are considered: deficit, expansion and center, where wake deficit is a measure of the decrease in downwind
15    wind speed, wake expansion describes the size of the downwind area affected by the wake and wake center defines the lateral position (meandering) of the wake area. Expressions for wake deficit, center and expansion were developed in (Frandsen et al., 2006; Jensen, 1983). To illustrate this, a small wind farm comprising of five turbines is shown in the layout of Fig. 7. Figure 8 shows a snapshot of the wind field where the wake effects are observed.

## 3 Variable speed control strategy

The so-called variable-speed operation is of particular interest for this concept because by removing the individual generators and power electronics from the turbines, the hydraulic drives need to replace the control actions to obtain the variable-speed functionality.

### 3.1 Pump controller

As shown in Eq. (7), it is possible to manipulate the transmitted torque of the pump using two different control degrees of freedom (in contrast with the electro-magnetic torque in a conventional turbine): the volumetric displacement of the pump and/or the pressure across it. In this case, the volumetric displacement of the pump from each turbine is controlled under a relatively constant pressure supply. Hence, the rotational speed of each rotor is able to be modified independently according to the local wind speed conditions. A constant pressure in the hydraulic network is desired, not only to keep the Pelton turbine operating at maximum efficiency as described in section. 2.4, but to be able to connect the water pumps from the individual turbines to the hydraulic network. In addition, maintaining a constant pressure supply is beneficial in minimizing fatigue damage to the hydraulic system components. This strategy is commonly known in hydraulic systems as 'secondary control' (Murrenhoff, 1999). The required volumetric displacement of the pump $e_{dem}$ is shown in Eq. (21) as a function of the measured rotational speed of the rotor $\omega_{r,meas}$ and the measured pressure at the pump location $\Delta p_{p,meas}$. The reference torque $\tau_{ref}$ is obtained from the steady-state torque-speed curves defined for different operating regions as in conventional variable-speed control strategies.

$$e_{dem} = \frac{\tau_{ref}\left(\omega_{r,meas}\right) - B_p \omega_{r,meas}}{V_p \left(1 + C_f\right) \Delta p_{p,meas}} \tag{21}$$

A first order low pass filter on the pressure measurement is employed to prevent actuation from the fluid transient fluctuations in the hydraulic network with the following transfer function form:

$$LPF\left(s\right) = \frac{1}{1 + \frac{s}{\omega_c}} \tag{22}$$

where the cut-off frequency $\omega_c$ was set at $16 \cdot 2\pi \ \mathrm{rad \ s^{-1}}$.

### 3.2 Spear valve controller

In order to achieve a constant pressure in the hydraulic network, linear actuation of the spear valve is used to constrict or release the flow rate through the nozzle area. The pressure control is based on a PI feedback controller and a cascade controller compensation to modify the linear position of the spear valve. A schematic of the proposed controller is shown in Fig. 9. Another option is to implement a constant pressure control as proposed in (Buhagiar et al., 2016), where a feedback controller is used in combination with feed-forward compensation.

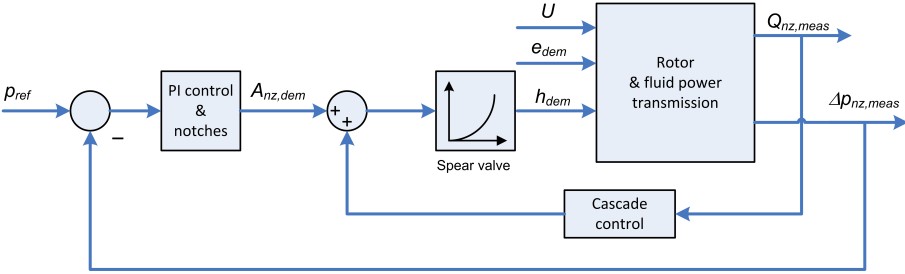

**Figure 9.** Pressure control schematic based on the spear valve position of the nozzle.

The PI controller is augmented with a second order low pass filter and a series of notch filters. A schematic showing the structure of the augmented controller is shown in Fig. 10. For the presented case studies, two notch filters are required to prevent excitation from the first two low damped modes of the hydraulic network located at $0.7 \cdot 2\pi$ rad s$^{-1}$ and $1.4 \cdot 2\pi$ rad s$^{-1}$ respectively.

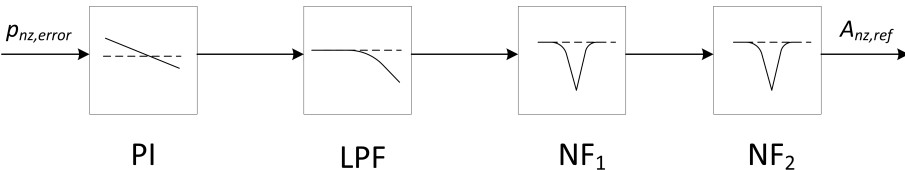

**Figure 10.** Schematic overview of the structure of the controller. The control blocks from left to right: Proportional-integral (PI), low-pass filter (LPF), notch filter 1 (NF1), notch filter 2 (NF2).

5    The low pass filter and the notch filters are described in the frequency domain according to Eqs. 23 and 24. The values of the different control parameters are displayed in Table 1. The negative values of the proportional and integral gain show that if the reference pressure is higher than the measured pressure at the nozzle (positive error input to the controller), the controller action should reduce the nozzle area to constrict the flow rate and induce a higher pressure. This inverse relation is reflected in the negative values of the controller gains.

$$LPF\left(s\right) = \frac{\omega_{LPF}^{2}}{s^{2} + 2\omega_{LPF}\,\zeta_{LPF}s + \omega_{LPF}^{2}} \tag{23}$$

$$NF_{i}\left(s\right) = \frac{s^{2} + 2\omega_{ni}\,\zeta_{ni}s + \omega_{ni}^{2}}{s^{2} + 2\omega_{ni}\,\beta_{ni}s + \omega_{ni}^{2}} \tag{24}$$

### 3.3  Pitch control

Above rated wind speed, the rated rotor speed is maintained by pitching collectively the rotor blades. A conventional PI pitch controller is proposed using the rotor speed error instead of the generator speed error. Due to the sensitivity of the aerodynamic

**Table 1.** Controller parameters of the spear valve augmented controller.

| Description | Symbol | value |
|---|---|---|
| Proportional gain | $K_P$ | $-2.7898 \cdot 10^{-10} \ \mathrm{m}^2 \ \mathrm{Pa}^{-1}$ |
| Integral gain | $K_I$ | $-1.0565 \cdot 10^{-10} \ \mathrm{m}^2 \ \mathrm{Pa}^{-1} \ \mathrm{s}$ |
| Low-pass filter frequency | $\omega_{LPF}$ | $1 \cdot 2\pi \ \mathrm{rad \ s}^{-1}$ |
| Low-pass filter parameter | $\zeta_{LPF}$ | 0.7 |
| Notch filter 1 frequency # 1 | $\omega_{n1}$ | $0.7 \cdot 2\pi \ \mathrm{rad \ s}^{-1}$ |
| Notch filter 2 frequency # 2 | $\omega_{n2}$ | $1.4 \cdot 2\pi \ \mathrm{rad \ s}^{-1}$ |
| Notch filter 1 parameter # 1 | $\zeta_{n1}$ | 0.01 |
| Notch filter 1 parameter # 2 | $\beta_{n1}$ | 0.7 |
| Notch filter 2 parameter # 1 | $\zeta_{n2}$ | 0.01 |
| Notch filter 2 parameter # 2 | $\beta_{n2}$ | 0.7 |

response of the rotor to the pitch angle, the value of the controller gains are modified as a function of the pitch angle through a gain-scheduled approach. The gain scheduled PI controller is shown in the next equations, where $K_{P/I}$ are the proportional and integral gains respectively, $K_{P/I,0}$ is the gain at rated pitch angle $\beta = 0$, and $\beta_K$ is the blade pitch angle at which the pitch sensitivity of aerodynamic power to rotor collective blade pitch has doubled from its value at the rated operating point.

$$\beta_{dem} = K_P(\beta) \, \omega_{r,error} + K_I(\beta) \int_0^t \omega_{r,error} \, dt \tag{25}$$

$$K_{P/I}(\beta) = K_{P/I,0} \, \frac{\beta_K}{\beta_K + \beta} \tag{26}$$

$$\omega_{r,error} = \omega_{r,rated} - \omega_{r,meas} \tag{27}$$

The values of the different gains are obtained in a similar way as described in (Jonkman et al., 2009), taking into account a modified apparent inertia at the low speed shaft and a transmission ratio which is set to one. To get rid of high frequency excitation, a low pass filter on the rotor speed measurement is used to prevent high frequency pitch action.

## 4 Simulation example

### 4.1 Wind farm conditions

The model described in the previous sections is used to assess the performance and operating conditions of a small hydraulic wind farm under specific wind conditions. Five turbines of 5MW each are interconnected, through a hydraulic network, to a 25MW Pelton turbine located at an offshore platform within $1 \ \mathrm{km}$ distance from the individual turbines. Two different wind speeds corresponding to below and above rated conditions are simulated. First, a wind field with a mean wind speed of $9 \ \mathrm{m/s}$

and 10% turbulence intensity (TI) is taken as the inflow condition during $1000s$. For above rated conditions, a mean wind speed of $15$ m/s and 12% TI is employed. The main parameters are shown in Table 2.

**Table 2.** Main design parameters for the offshore wind turbine with fluid power transmission.

| Design parameter | | Design parameter | |
|---|---|---|---|
| Rotor diameter | 126 m | Drivetrain concept | Hydraulic |
| Rated wind speed | 11.4 m/s | Nominal water pressure | 150 bar |
| Design tip speed ratio $\lambda$ | 7.55 | Pump volumetric disp | 10.2 L/rpm |
| Max power coefficient $C_P$ | 0.485 | Lines length | 1 km |
| Rated power | 5 MW | Lines diameter | 0.5 m |
| Max blade tip speed | 80 m/s | Nozzle nominal diameter | 43.2 mm |

The results from the simulations are compared with those of a reference wind farm comprising of 5MW NREL turbines (Jonkman et al., 2009), using the same wind farm layout and environmental conditions. A schematic of the individual turbines and configurations used in the simulation example for both wind farms is shown in Fig. 11. The capital letters A, B and C are used as a reference to present the results at specific points. For the hydraulic turbines, a separate boost pump is required to supply the water to the pump located at the nacelle. Together with the filters and cooling system these components comprise the auxiliary equipment which is not included in the analysis. The same consideration is made in the conventional wind turbine technology where the lubrication, filtering and cooling power required by the gearbox and generator is not included in the analysis.

## 4.2 Time-domain results

The results of the time domain simulations are presented in terms of the main operational parameters such as mechanical power, rotor speed and pitch angle for the five turbines. For below rated conditions Fig. 12 shows the transient response of the reference and the hydraulic wind farm. The results demonstrate that for the considered scenario and with the current control strategy, the hydraulic wind farm is able to generate electricity from the pressurized water flow to the central platform via a Pelton turbine. In terms of performance it is observed that the turbines in the hydraulic wind farm show larger fluctuations of the rotor speed in comparison with the reference case; this effect is also reflected in the increased pitch action required for the same wind speed conditions. A possible explanation of the more pronounced changes of the rotor speed is that the resulting torque demand generated by the hydraulic system is slower than in the reference case due to the higher fluid inertia of the hydraulic network. From a reliability point of view, the increased pitch action might have an important consequence on the life time of the pitch system. During the first 100s, the hydraulic wind farm shows high frequency fluctuations in the pressure and, consequently, in the total power output of the array. These higher fluctuations are due to the initial conditions of the pressure control settings in combination with the high fluid inertia in the hydraulic network. The changes in pressure and volumetric

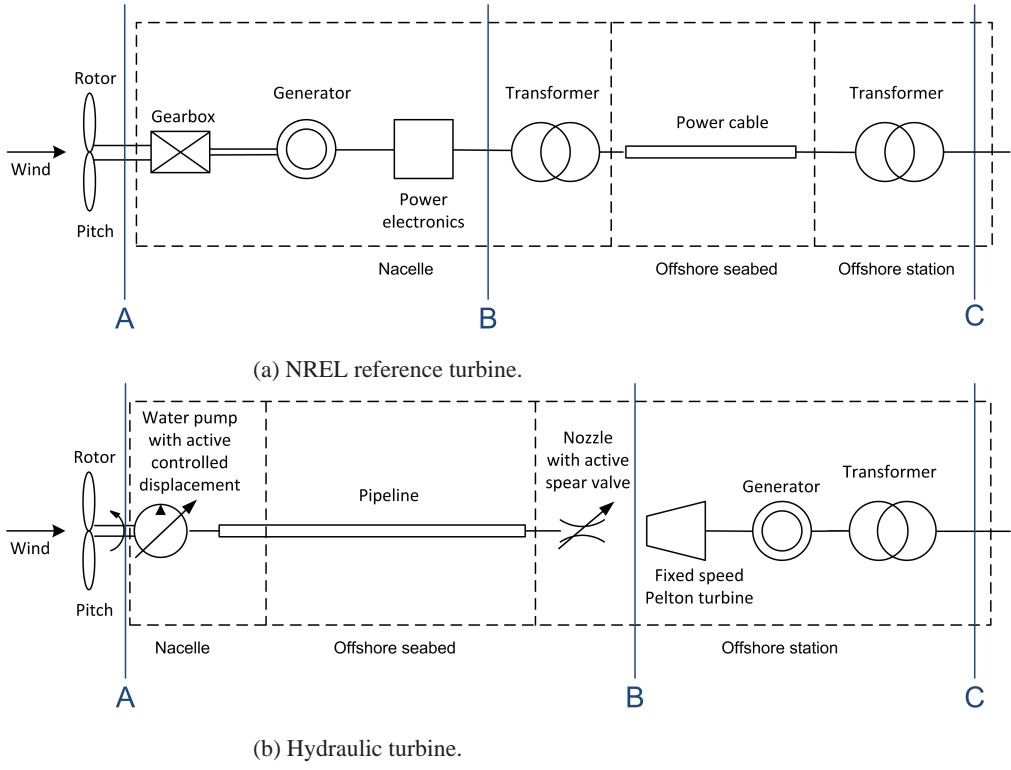

(a) NREL reference turbine.

(b) Hydraulic turbine.

**Figure 11.** Simplified schematic with the main components involving the energy conversion for a reference offshore wind turbine and the proposed hydraulic concept.

flow rate at the nozzle, have small influence on the efficiency of the Pelton turbine, which is maintained relatively constant and well above 90% during the whole simulation time, except for the first 100s of transient conditions.

For above rated conditions, the simulation results are shown in Fig. 13. It is observed that both concepts are able to keep the rotor speed operating within a constant speed band while producing relatively constant power. Likewise, the pitch actuation is very similar in both wind farms, which is not unexpected since the same pitch controller is used. Once more, the transient operation in the electrical power production is more pronounced in the case of the hydraulic wind farm because of the high hydraulic inertia of the hydraulic network. High frequency oscillations are observed in the electrical power as a consequence of the pressure waves travelling along the network.

**Table 3.** Performance overview of time domain results for below rated conditions.

| | Averaged power [MW] | | | | | | Efficiency [-] | | |
| --- | --- | --- | --- | --- | --- | --- | --- | --- | --- |
| | Mechanical point A | | Transmitted point B | | Electrical point C | | Power coeff $C_P$ | A to B $\eta_{AB}$ | B to C $\eta_{BC}$ |
| Wind farm concept | | | | | | | | | |
| NREL reference | mean | std | mean | std | mean | std | mean | mean | mean |
| WT1 | 3.12 | 0.86 | 2.95 | 0.81 | 2.61 | 0.72 | 0.483 | 0.944 | 0.885 |
| WT2 | 2.23 | 0.60 | 2.11 | 0.57 | 1.87 | 0.50 | 0.483 | 0.944 | 0.885 |
| WT3 | 2.90 | 0.88 | 2.74 | 0.83 | 2.42 | 0.73 | 0.483 | 0.944 | 0.885 |
| WT4 | 2.99 | 0.83 | 2.82 | 0.78 | 2.50 | 0.69 | 0.483 | 0.944 | 0.885 |
| WT5 | 2.10 | 0.58 | 1.98 | 0.54 | 1.75 | 0.48 | 0.483 | 0.944 | 0.885 |
| Total | 13.3 | | 12.6 | | 11.1 | 1.90 | - | - | - |
| Hydraulic with pressure control | | | | | | | | | |
| WT1 | 3.06 | 0.92 | - | - | - | - | 0.479 | - | - |
| WT2 | 2.22 | 0.69 | - | - | - | - | 0.482 | - | - |
| WT3 | 2.84 | 0.90 | - | - | - | - | 0.479 | - | - |
| WT4 | 2.94 | 0.89 | - | - | - | - | 0.480 | - | - |
| WT5 | 2.08 | 0.65 | - | - | - | - | 0.482 | - | - |
| Total | 13.1 | | 11.6 | 2.58 | 10.2 | 2.71 | - | 0.88 | 0.877 |

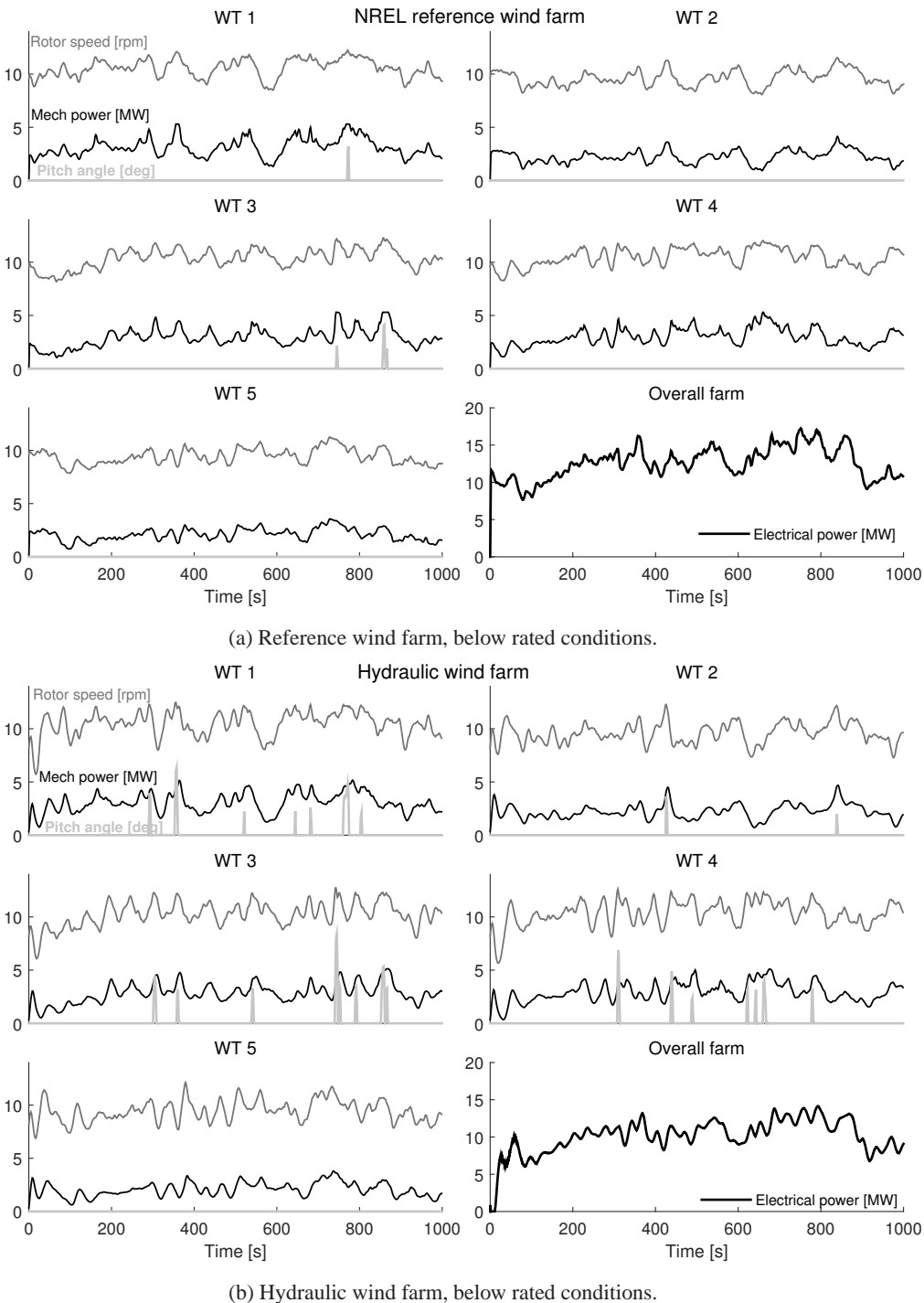

(a) Reference wind farm, below rated conditions.

(b) Hydraulic wind farm, below rated conditions.

**Figure 12.** Time domain results for a wind farm comprising of 5 turbines subject to a wind field with a mean speed of $9$ m/s and $10\%$ turbulence intensity.

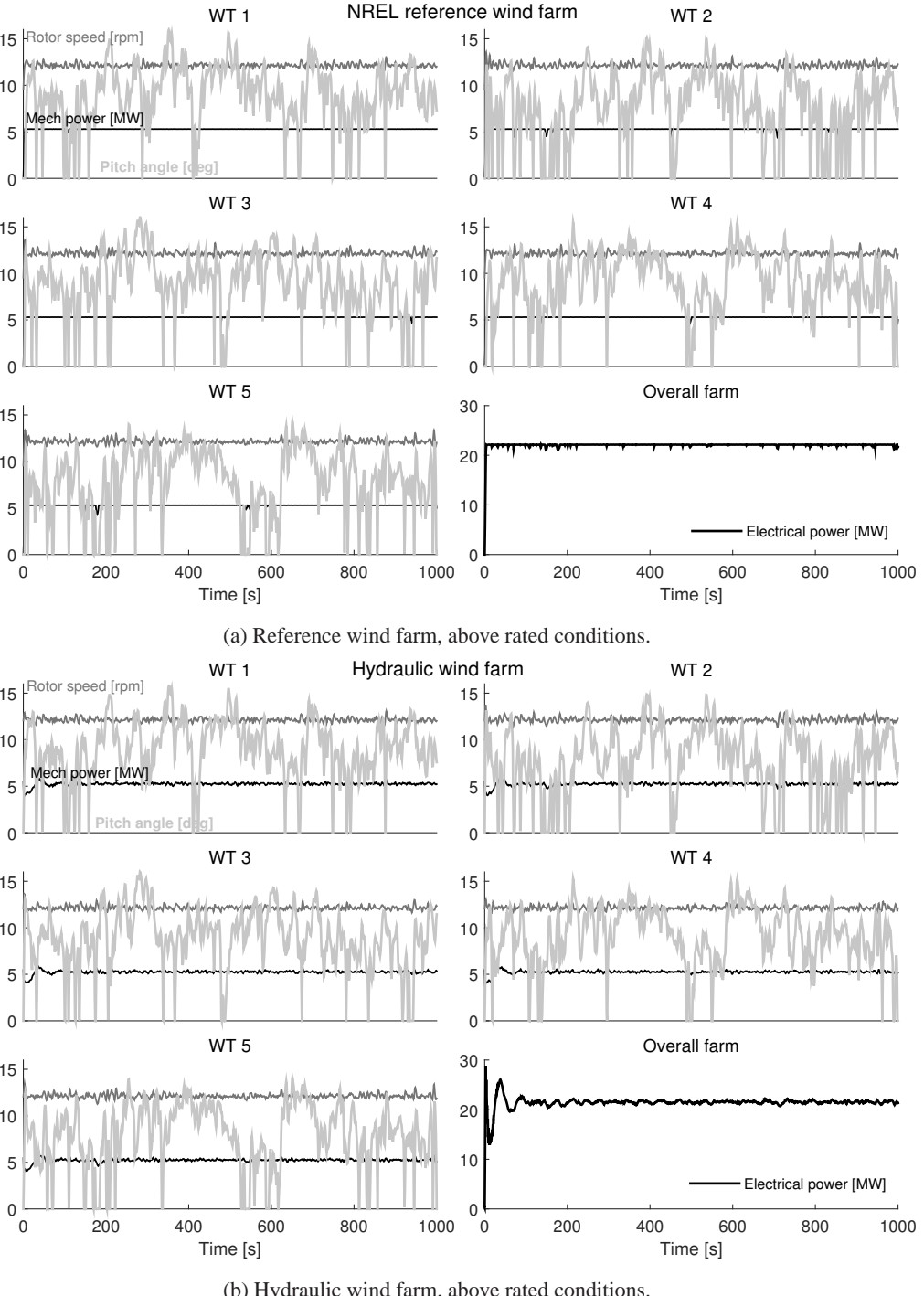

(a) Reference wind farm, above rated conditions.

(b) Hydraulic wind farm, above rated conditions.

**Figure 13.** Time domain results for a wind farm comprising of 5 turbines subject to a wind field with a mean speed of $15$ m/s and $12\%$ turbulence intensity.

## 4.3 Performance comparison

The performance of both wind farms for the considered conditions is summarized in the bar charts of Figs. 14 and 15 where the averaged values with the standard deviation of the power transmission and conversion are displayed. The numerical values together with the averaged efficiencies are summarized in Tables 3 and 4.

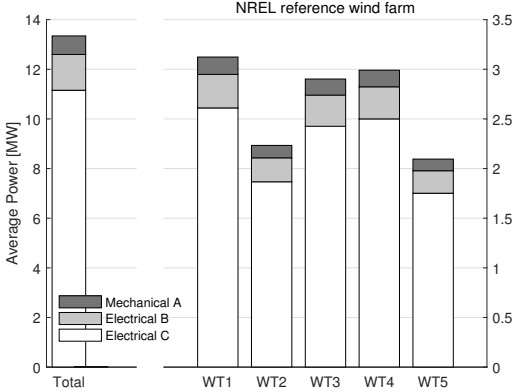

**Figure 14.** Power performance for the reference wind farm, below rated conditions.

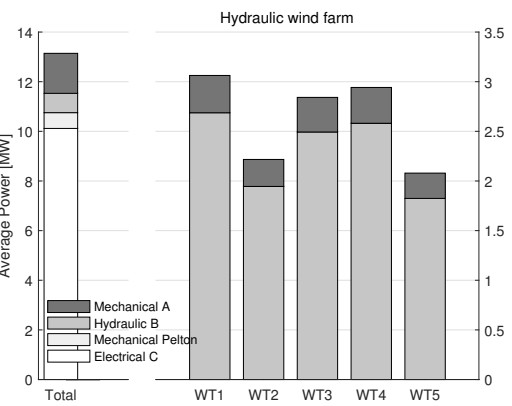

**Figure 15.** Power performance for the hydraulic wind farm, below rated condition.

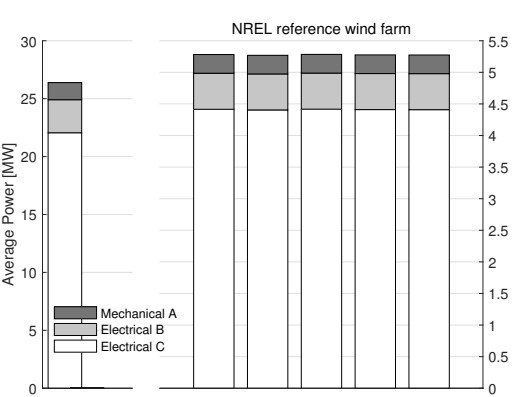

**Figure 16.** Power performance for the reference wind farm, above rated conditions.

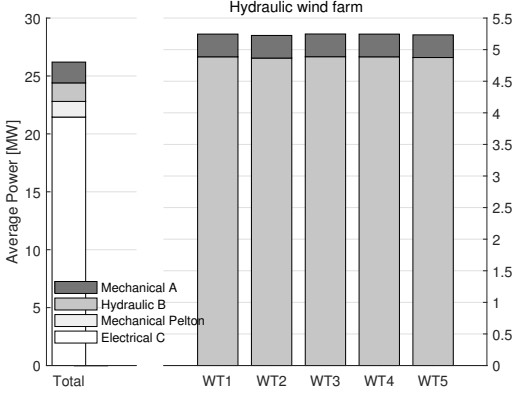

**Figure 17.** Power performance for the hydraulic wind farm, above rated conditions.

5    The first observation based on the general results for both wind farms is the reduced power performance of turbines WT2 and WT5. The performance of these two turbines is directly affected by the generated wake from turbines WT1 and WT4. In contrast, turbines WT1, WT3 and WT4 are not affected by any other wake interaction.

After including the performances of the main subsystems involved in the conversion and transmission of wind energy in a wind farm, the results show that the overall efficiency of a hydraulic wind farm is lower for a hydraulic concept compared

**Table 4.** Performance overview of time domain results for above rated conditions.

| Wind farm concept | Averaged power [MW] | | | | | | Efficiency [-] | | |
| | Mechanical point A | | Transmitted point B | | Electrical point C | | Power coeff $C_P$ | A to B $\eta_{AB}$ | B to C $\eta_{BC}$ |
| NREL reference | mean | std | mean | std | mean | std | mean | mean | mean |
|---|---|---|---|---|---|---|---|---|---|
| WT1 | 5.28 | 0.22 | 4.99 | 0.21 | 4.41 | 0.18 | 0.249 | 0.944 | 0.885 |
| WT2 | 5.27 | 0.23 | 4.97 | 0.22 | 4.40 | 0.19 | 0.284 | 0.944 | 0.885 |
| WT3 | 5.28 | 0.22 | 4.99 | 0.21 | 4.42 | 0.18 | 0.251 | 0.944 | 0.885 |
| WT4 | 5.28 | 0.23 | 4.98 | 0.22 | 4.41 | 0.19 | 0.244 | 0.944 | 0.885 |
| WT5 | 5.27 | 0.23 | 4.98 | 0.22 | 4.41 | 0.19 | 0.277 | 0.944 | 0.885 |
| Total | 26.4 | | 24.9 | | 22.1 | 0.92 | - | - | - |
| Hydraulic with pressure control | | | | | | | | | |
| WT1 | 5.24 | 0.18 | - | - | - | - | 0.247 | - | - |
| WT2 | 5.22 | 0.19 | - | - | - | - | 0.282 | - | - |
| WT3 | 5.25 | 0.18 | - | - | - | - | 0.250 | - | - |
| WT4 | 5.25 | 0.18 | - | - | - | - | 0.243 | - | - |
| WT5 | 5.23 | 0.19 | - | - | - | - | 0.274 | - | - |
| Total | 26.2 | | 24.4 | 1.40 | 21.4 | 1.44 | - | 0.931 | 0.87 |

to conventional technology. For the presented operating conditions the hydraulic wind farm overall efficiency was between $0.772 - 0.810$ compared to $0.835$ excluding aerodynamic performance. The most important losses in the hydraulic concept are attributed to the variable displacement pumps and friction losses in the hydraulic network. Despite having a slower response due to high water inertia, the hydraulic concept also showed higher standard deviations in the generated electrical power due
to pressure transients in the hydraulic network.

## 4.4   Full stop of turbines in the hydraulic wind farm

In the proposed hydraulic wind farm, all turbines are coupled to the same hydraulic network. This means that the pressure response in the hydraulic network is influenced by the individual flow rates of each turbine water pump. At the same time, the transmitted torque to each rotor is influenced by the local pressure at the water pumps. When abrupt changes in flow or
pressure are induced as a result of either accidental or normal operation, pressure transients in the form of traveling waves are introduced in the hydraulic network which have to be taken into account. Furthermore, with the 'secondary control' strategy proposed for the hydraulic system, the main large system effect of having several turbines connected to the hydraulic network is mostly determined by the ability of the spear valve and its controller to keep a constant pressure in the system. From this perspective, if one or more turbines are brought to a full stop, the spear valve should be able to maintain a relatively constant
pressure in order for the remaining turbines to keep operating within design limits.

The following simulation presents the results of the scenario in which two turbines are brought to a full stop at different moments of time. Starting from the same environmental wind conditions from the previous example, above rated conditions with mean speed of 15 m/s and 12% turbulence intensity, the first turbine WT1 shuts down at 200s followed by the second turbine WT4 at 600s.

The operational parameters of each turbine including the full stop of WT1 and WT4 are shown in Fig. 18. It is observed that the overall electrical power of the hydraulic wind farm is also decreased every time a turbine is stopped. As a consequence of each sudden stop of the flow rate, the decrease in power is accompanied by a negative overshoot which is directly related to the transient response of the pressure across the nozzle and its effect on the Pelton efficency.

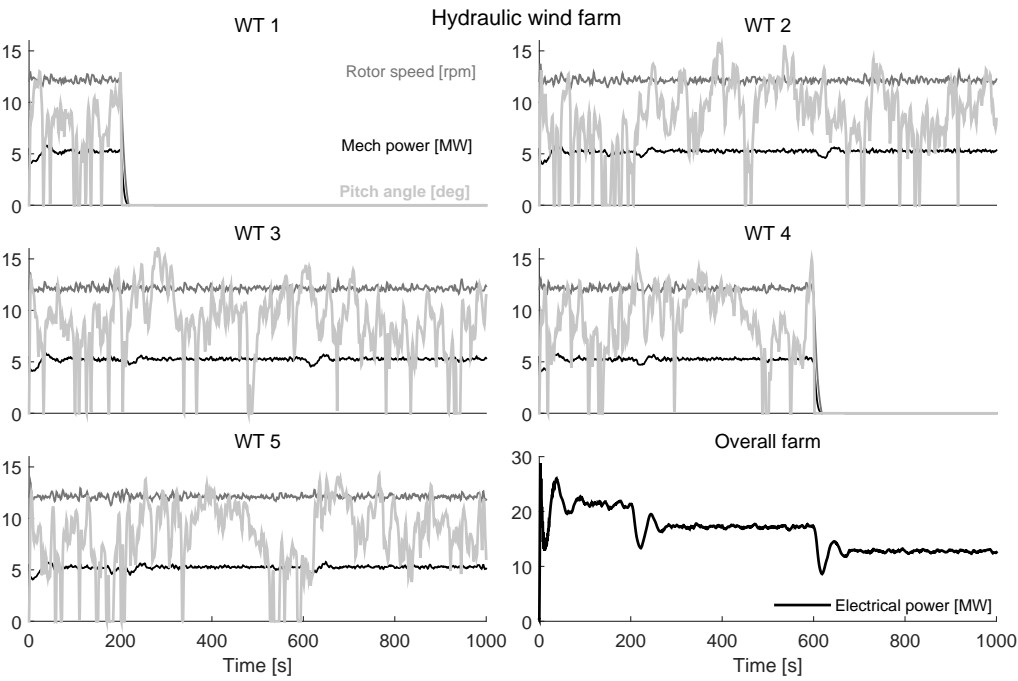

**Figure 18.** Time domain results for a hydraulic wind farm subject to a wind field with a mean speed of $15$ m/s and $12\%$ turbulence intensity and full stop of turbines WT1 and WT4 at 200s and 600s respectively.

In order to compensate for the overall decrease in flow rate through the hydraulic network, each of the water pumps from
the remaining operating turbines are required to increase their volumetric displacement as observed in the normalized control signal $e_{dem}$ in Fig. 19. For each turbine full stop, it is observed that the negative overshoot in the pressure difference across the nozzle has the same magnitude but the resulting pressure difference is lower. Thus, the efficiency of the Pelton turbine is affected in a different manner depending on the value of the pressure difference. For a fixed-speed Pelton turbine, the lower pressures and consequently lower jet velocities results in increased runner speed ratios which have a direct impact on the Pelton
conversion efficiency.

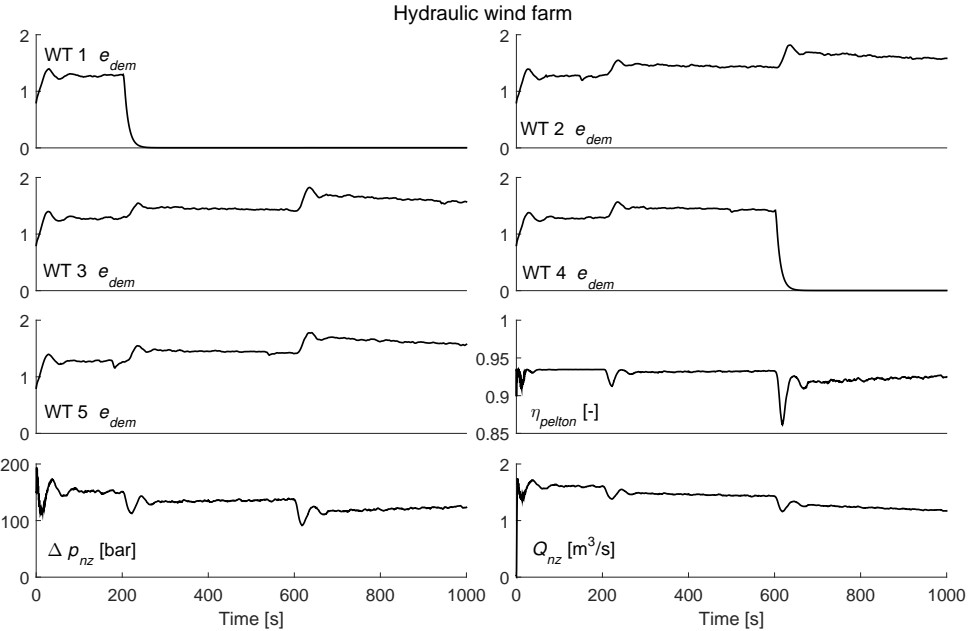

**Figure 19.** Operating parameters of a hydraulic wind farm subject to a wind field with a mean speed of $15\ \mathrm{m/s}$ and $12\%$ turbulence intensity and full stop of turbines WT1 and WT4 at 200s and 600s respectively.

## 5  Conclusions

The numerical model of a hydraulic wind power plant aimed to generate electricity in a centralized manner has been presented. The model demonstrates that on the basis of physical principles, it is possible to centralize electricity generation by dedicating the individual turbines inside a wind farm to pressurize water into a hydraulic network and then use the pressurized flow in a 5  Pelton turbine. A variable speed operation of the turbine is proposed in combination with a pressure controller in the nozzle spear valve to avoid the excitation of flow and pressure dynamics in the hydraulic network. Furthermore, the constant pressure system allows to include a fixed-speed Pelton turbine which simplifies the integration with the electrical grid.

Despite the stochastic turbulent wind conditions and wake effects, the results of the presented case studies indicate that the individual wind turbines are able to operate within operational limits for both below and above rated wind conditions. 10  Compared to a reference wind farm based on conventional wind turbine generator technology, the hydraulic collection and transmission has a lower efficiency due to the losses induced by the variable displacement water pumps and friction losses in the hydraulic network. The continuous operation of the hydraulic wind farm has been shown by bringing two different turbines to a full stop in above rated wind conditions. Further work includes the evaluation of alternative control strategies to assist the performance evaluation of the proposed centralized electricity generation approach. Other prospects of the hydraulic concept 15  include the development and integration of an energy storage system using hydraulic accumulators. It is expected that these hydraulic devices will minimize the electrical power fluctuations for turbulent wind conditions.

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
