# Peer review of "Simulation of an offshore wind farm using fluid power for centralized electricity generation"

_Wind Energy Science, 2016_

## Referee Comment (RC1) · M Muskulus (Referee) · 24 Jan 2017

General comments

This paper is a solid simulation study of a wind farm with a hydraulic network and single generator for power take-off. The system is modeled by ordinary differential equations for the hydraulic network and the rotor dynamics, including the wind turbine controller. Aerodynamic loads on the turbines are obtained from turbulent wind fields with a one-dimensional model based on steady state torque coefficients. Wake effects in the wind farm seem to have been included using standard models. Compared to previous studies, the novely seems to be the consideration of a complete wind farm, instead of a single turbine.

The topic is relevant, and the paper is well written. I have a few comments on improving

aspects of the presentation, see below. Apart from that, it should be better described what the novelty is compared to earlier studies, and the conclusions should contain more information on what new knowledge we can learn from this study.

I would like to recommend the paper for publication, but have one major objection. From what I can see, the paper has been invited for the Special Issue of Wind Energy Science from the Science of Making Torque (TORQUE2016) conference. This should probably be mentioned somewhere in the paper, e.g. in a footnote. More importantly, the paper submitted here is, as far as I can tell, completely identical to the paper submitted at TORQUE2016. This is of course in conflict with policies on authorship at Wind Energy Science that ask for at least 40 percent new content. It is also ethically problematic, as it basically means twice the voluntary review and editorial effort, without any new value to the scientific community. Now, I will give the author the benefit of the doubt and will assume that he was not properly aware of these issues when asked to submit his paper for the Special Issue. Nevertheless, without additional results of interest, the paper cannot be published. One of my comments below contains a suggestion.

Specific comments

1. The paper should probably mention (e.g. in a footnote) that it is an extended and updated version of a paper previously presented at TORQUE2016 conference, and published in IOP Journal of Physics: Conference Series.

2. Continuing from the previous item, the paper needs to contain at least 40 percent new content, which is currently not the case.

3. Introduction: "This paper continues with previous work" - It would help the reader if the scope and achievements of the previous work were briefly reported. That way the research is placed more into context, and it becomes easier to evaluate what is new here.

4. Are there any system effects when running the concept with more than one turbine? The performance and results obtained for the wind farm should be compared (in a meaningful way) with results for a single turbine.

5. The concept is based on the use of seawater. I assume that corrosion becomes an important issue then. Does the author have some comments for the readers on this?

6. As it is proposed to use only one turbine and generator, reliability of these becomes a critical issue. Has the author any thought on this that he would like to share with the readers?

7. Eqs. 6-7: The notation is slightly confusing. I assume that $V(e)$ is a function depending on the variable $e$, later shown in Eq. 8. However, also other terms in Eqs. 6-7 are functions that depend on parameters. To be consistent, I suggest that you simply use $V$ in Eqs. 6-7 and clarify $V(e) = eV_{p,\text{max}}$ in Eq. 8.

8. Section 2.1.3: The pitch actuator model is based on a proportional regulator. Why not also a derivative or integrator component? Why is the pitch actuator model needed?

9. Section 2.3: The nozzle length $L_{nz}$ should be indicated in Figure 4 as well.

10. Section 2.4: What is the value of the vena contracta coefficient used here?

11. Section 3.1: "A low pass filter on the pressure measured is employed" - What are the filter characteristics?

Technical corrections (p=page, l=line)

- p1, l16: "and the electricity is then ..."?

- p3, l10: "defined as the ratio of the tangential velocity ..."?

- Section 4.2: "experience higher excursions of the rotor speed" - maybe change to "show larger fluctuations of the rotor speed"?

---

## Author Comment (AC1) · 1 Feb 2017

Thank you to the reviewer for the comments on the paper. Indeed this submitted paper is an extended version of the work presented in the Science of Making Torque (TORQUE2016) conference. The author would like to emphasize that the content is not completely identical to the conference paper as the reviewer suggest. Following the received invitation where it was stated to "Prepare an expanded version of your TORQUE paper, indicatively containing 30-40% additional original material (in the form of a more detailed or expanded description of theory and/or methods, additional results, etc.)" the following overview highlights the additions and new content to this work (as indicated with the text in bold):

1 Introduction - Added conceptual comparison between a conventional and the proposed offshore wind farm in Figure 1.

2 Wind farm model overview

- 2.1 Wind Turbines
- 2.1.1 Aerodynamic model
- 2.1.2 Hydraulic drive train model
- 2.1.3 Pitch actuator model Added
- 2.1.4 Structural model Added
- 2.2 Hydraulic network Expanded with the addition of a schematic, see Figure 3
- 2.3 Nozzle and spear valve A schematic of the spear valve was added, see Figure 4.
- A graph showing the nozzle characteristic is added and presented in Figure 5.
- 2.4 Pelton turbine
- 2.5 Environmental conditions
- 3 Variable speed control strategy
- 3.1 Pump controller

3.2 Spear valve controller – Expanded with the details of the controller and filters, a schematic was added, see Figure 9

- 3.3 Pitch control Added
- 4 Simulation example 4.1 Wind farm conditions
- 4.2 Time domain results
- 4.3 Performance comparison
- 5 Conclusions

The author acknowledge that no additional results were included up to this point, however all the added material is considered by the author to be of importance in the overall structure and completeness of the work and was not possible to be included in the conference paper due to page limitations. If consider necessary for publication, the author could address this issue by the inclusion of an extra case scenario which could add insight on the behaviour of the proposed model but most likely the current conclusions and findings will remain. New results and figures will be added in the revised version.

The specific replies to the reviewer comments [RC1] are addressed by the author [AC] in the following lines:

1. [RC1] "The paper should probably mention (e.g. in a footnote) that it is an extended and updated version of a paper previously presented at TORQUE2016 conference, and published in IOP Journal of Physics: Conference Series."

[AC] As this paper was invited for the special Issue on The Science of Making Torque from Wind (TORQUE) 2016, this was not considered necessary. The author will comply to the editors instructions.

2. [RC1] "Continuing from the previous item, the paper needs to contain at least 40 percent new content, which is currently not the case."

[AC] See general comment at the beginning of this document .

3. [RC1] "Introduction: 'This paper continues with previous work' - It would help the reader if the scope and achievements of the previous work were briefly reported. That way the research is placed more into context, and it becomes easier to evaluate what is new here."

[AC] The author agrees. The scope and achievements of previous work were in the context of a single turbine, while this paper considers a complete wind farm. The manuscript will be modified accordingly.

4. [RC1] "Are there any system effects when running the concept with more than one

СЗ

turbine? The performance and results obtained for the wind farm should be compared (in a meaningful way) with results for a single turbine.

[AC] The main difference with respect to a single turbine is that in a wind farm all the turbines are coupled to the hydraulic network. This means that the pressure response in the hydraulic network is influenced by the individual flow rate of each turbine water pump and at the same time the transmitted torque to each turbine is influenced by the pressure at the water pumps. When abrupt changes in flow or pressure are induced as a result of either accidental or normal operation, pressure waves are introduced in the hydraulic network which have to be taken into account. With the 'secondary control' strategy proposed for the hydraulic system, the main large system effect of having several turbines connected to the hydraulic network is mostly determined by the ability of the spear valve and its controller to keep a constant pressure in the system. Provided that a relatively constant pressure is maintained, each turbine will be able to operate independently.

5. [RC1] "The concept is based on the use of seawater. I assume that corrosion becomes an important issue then. Does the author have some comments for the readers on this? "

[AC] In the proposed concept, an open-loop circuit is considered (i.e. the fluid is not circulating) with seawater as hydraulic fluid. The choice of seawater as hydraulic fluid is preferred because of its availability and environmental friendly nature when compared to oil hydraulics. It is important to consider that seawater contains a high concentration of minerals, which give it a high degree of hardness. It also contains dissolved gases such as oxygen and chlorine which cause corrosion. Despite its corrosive nature, the use of seawater hydraulics has already been used in some industrial applications, where in terms of safety, water hydraulics might be preferred due to potential fire hazards or risk of leakage as is the case of the mining industry. An example in the offshore industry includes the seawater hydraulic system for deep sea pile driving incorporating high pressure water pumps (IHC Hydrohammer, 2009)[1]. A key advantage of this system is that the use of an open loop circuit cancels the need for cooling equipment, a disadvantage is that it is likely that filters have to be cleaned more frequently.

The description above will be added to the manuscript.

[1] M. Schaap. Seawater Driven Piling Hammer. In Proceedings of the Dutch Fluid Power Conference in Ede, September 2012. (reference added)

6. [RC1] "As it is proposed to use only one turbine and generator, reliability of these becomes a critical issue. Has the author any thought on this that he would like to share with the readers?"

[AC] Indeed, by using only one or a few turbines and generators, the reliability of these components become an important aspect. Modern hydro-turbines have been developed with typical capacities of 500 MW operating for decades with enough operational and maintenance experience gained from conventional hydro-power plants. On the other hand using hydro turbines in combination with renewable energy sources such as offshore wind energy has not been explored. The concept itself is still in predevelopment phase and therefore there is a lack of real data supporting the reliability. It is also expected that by having the whole electrical generation equipment in one offshore central platform instead of having it in a constraint space hundred meters above sea level, would have a positive impact regarding O&M costs.

7. [RC1] "Eqs. 6-7: The notation is slightly confusing. I assume that V(e) is a function depending on the variable e, later shown in Eq. 8. However, also other terms in Eqs. 6-7 are functions that depend on parameters. To be consistent, I suggest that you simply use V in Eqs. 6-7 and clarify V(e) = eVp;max in Eq. 8."

[AC] The author agrees to modify Eqns 6-7 to include the reviewer's comment.

8. [RC1] "Section 2.1.3: The pitch actuator model is based on a proportional regulator. Why not also a derivative or integrator component? Why is the pitch actuator model needed?"

[AC] The pitch actuator model is needed to account for any blade-pitch actuator dynamic effects. This means the slow or fast response of the pitching mechanism to the control command signal. The derivative or integrator components are considered to be included in the pitch control which is in series with the pitch actuator, see Section 3.3.

9. [RC1] "Section 2.3: The nozzle length Lnz should be indicated in Figure 4 as well."

[AC] The author agrees, Figure 4 will be modified accordingly.

10. [RC1] "Section 2.4: What is the value of the vena contracta coefficient used here?"

[AC] A value of Cv=0.99 was used according to (Thake, 2000)[2]. Please note that the vena contracta phenomenon does not influence the nozzle efficiency.

[2]Thake, J. The Micro-hydro Pelton Turbine Manual. Practical Action Publishing, 2000. (reference added)

11. [RC1] "Section 3.1: 'A low pass filter on the pressure measured is employed' What are the filter characteristics?"

[AC] A first order low pass filter was used with the following transfer function form:

LPF(s) = 1/(1+s/wc)

where the cut-off frequency wc was set at 32 pi [rad s-1]. This description will be included in the manuscript. A new table adding the controller parameters of the augmented controller from Eqns 22 and 23 will also be included.

All the technical corrections will be incorporated in the text. The author hopes that the proposed modifications to the manuscript and replies to the reviewers satisfy your requests.

---

## Referee Comment (RC2) · T. Sant (Referee) · 27 Feb 2017

**General Comments**

This paper deals with a new concept for offshore wind power conversation based on hydraulic transmission. The generator is replaced by a positive-displacement pump that is directly coupled to the wind turbine rotor. Each wind turbine supplies a pressurised source of sea water to a centralised hydro-electric station consisting of a Pelton wheel coupled to a synchronous machine. This study builds on previous work and describes numerical models to simulate the energy conversion processes, taking into account for the unsteady effects in the hydraulic pipeline network. The overall energy conversion efficiency of a typical hydraulic wind farm is compared with that of a conventional farm based on wind turbine generator technology. The paper is well-structured. Use of the

English language is very good. The title reflects the general content of the paper. The following are specific comments that have to be addressed for the paper to be suitable for publication

Specific Comments 1. Introduction, Page 2: the paper should present a more detailed overview of work carried out in this area so far and what new work is being presented in this study

2. Page 2, line 8: amend sentence to end as follows: "where the results are compared with those of a typical wind farm based on conventional wind turbine generator technology.

3. Page 3: Equation (3) is missing. The equation number (3) is being indicated.

4. Page 3, line 15: add a fullstop – "...both the rotor and support structure. Their effects on the..."

5. Page 3, line 16: remove the coma "..degree of freedom will absorb."

6. Page 3, line 21: add full stop – "..as a first order differential equation. The mass moment..."

7. Page 3, line 25: "is obtained for each rotor revolution."

8. Page 4, Figure 2 caption: "Subsystem block diagram of a single turbine..."

9. Page 4, line 9: remove coma – "displacement of the pump are approximated by a  $\dots$ "

10. Page 4, Eqt (12): Section 2.1.4 should include a brief explanation of how eqt (2) is used in conjunction with eqts (1,2) to determine the rotor torque.

11. Page 5, section 2.2, line 17: it should be clarified in the text that linearity only holds for laminar flows. For turbulent flow, the non-linear equations have to be applied.

12. Page 7, line 16: a more elaborate explanation is required about the fundamental

WESD
physics governing Pelton wheel operation: If the rpm is kept fixed, then the jet velocity and hence the press drop across the nozzle should be also fixed. k should also be fixed at the optimal value of 0.5 for optimal efficiency. An explanation of how this condition is applied in the numerical solution is necessary.

13. Page 9, first line: "so-called"

14. Page 9, line 9: this is linked to comment 12 above. Explain in the text why you have a constant pressure supply. To what extent is the control system able to maintain a constant pressure when intermittent wind conditions cause the water flowrate to change abruptly?

15. Page 9, section 3.2, first line: it is worth mentioning that maintaining a constant pressure supply is beneficial in minimising fatigue damage to the hydraulic system components.

16. Page 19, first line. Explain the difference between the control systems of Buhagiar et al and that being proposal here.

17. Page 11, line 9: Include a table with the derive values for the different gains

18. Page 12, line 15: wouldn't a compressed air or weighted accumulator help solve the problem of increased activity of the pitch controller?

19. Page 13, Figure 10: if the hydraulic turbine only includes an open-loop system with the pump housed at the nacelle, then a separate boost pump is required to be able to supply the sea water up the hub height. Has this been factored in the analysis?

20. Page 18, line 1: add a full stop – "conventional technology. For the presented..."

21. Page 18, line 3: quote here the percentage efficiency of the pump and that of the hydraulic network.

22. Page 18: Conclusions - comment on any opportunities for costs reduction offered by the new concept

WESD

---

## Author Comment (AC2) · 24 Apr 2017

Author response to reviewer #1

Thank you to the reviewer for the comments on the paper. The specific replies to the reviewer comments [RC1] are addressed by the author [AC] in the following lines with respect to the revised version:

1. [RC1] "The paper should probably mention (e.g. in a footnote) that it is an extended and updated version of a paper previously presented at TORQUE2016 conference, and published in IOP Journal of Physics: Conference Series."

[AC] As this paper was invited for the special Issue on The Science of Making Torque from Wind (TORQUE) 2016, this was not considered necessary following the example

of the other papers submitted for this special issue. The author will comply to the editors instructions.

2. [RC1] "Continuing from the previous item, the paper needs to contain at least 40 percent new content, which is currently not the case."

[AC] The revised version includes now a more detailed explanation of the motivation of the proposed concept together with opportunities for cost reduction. A expanded description of the different models employed, new figures and schematics are now included. The details and description of the controller are added with respect to the conference paper. New results are also presented in the form of an extra case scenario where the hydraulic wind farm is simulated while two turbines are brought to a full stop. The new results give better insight on the behaviour of the hydraulic model in particular with the proposed pressure controller (See also Figures 18 and 19). The conclusions also include further work. More references were also employed in the revised version

3. [RC1] "Introduction: 'This paper continues with previous work' - It would help the reader if the scope and achievements of the previous work were briefly reported. That way the research is placed more into context, and it becomes easier to evaluate what is new here."

[AC] The author agrees, similar comment was done by RC2. The following paragraph has been added: The modelling and analysis of a single turbine with hydraulic technology has been previously presented for variable-speed control strategies. Simulations of an individual turbine with an oil based hydrostatic transmission have been presented in (Jarquin Laguna et al., 2014). The results showed good dynamic behaviour for turbulent wind conditions where reduced fluctuations of the drivetrain torque and power are obtained despite the reduced energy capture. The integration of a single turbine with a Pelton runner using water hydraulics was introduced in (Jarquin Laguna, 2015), where a passive variable speed strategy was proposed. However, the addition and simulation of more turbines to the hydraulic network was not included. In an effort to
assess the trade-offs implied by the proposed hydraulic concept, this paper extends the time-domain simulations to evaluate the performance and operational parameters of five turbines coupled to a common hydraulic network for a hypothetical wind farm with centralized electricity generation. In the first part of this work, an overview of the wind farm model is presented together with the control strategy of the hydraulic components; the second part describes a case example where the results are compared with those of a typical wind farm based on conventional wind turbine generator technology.

4. [RC1] "Are there any system effects when running the concept with more than one turbine? The performance and results obtained for the wind farm should be compared (in a meaningful way) with results for a single turbine.

[AC] That is correct, the system effects are better illustrated in the extra case scenario included in the revised version where two turbines are brought to a full stop during above rated wind conditions, see description in section 4.4

5. [RC1] "The concept is based on the use of seawater. I assume that corrosion becomes an important issue then. Does the author have some comments for the readers on this? "

[AC] The following paragraph and reference has been added in the introduction: In the proposed concept, an open-loop circuit is considered (i.e. the fluid is not circulating) with seawater as hydraulic fluid. The choice of seawater as hydraulic fluid is preferred because of its availability and environmental friendly nature when compared to oil hydraulics. It is important to consider that seawater contains a high concentration of minerals, which give it a high degree of hardness. It also contains dissolved gases such as oxygen and chlorine which cause corrosion. Despite its corrosive nature, the use of seawater hydraulics has already been used in some industrial applications, where in terms of safety, water hydraulics might be preferred due to potential fire hazards or risk of leakage as is the case of the mining industry. An example in the offshore industry includes the seawater hydraulic system for deep sea pile driving incorporating high
pressure water pumps (Schaap 2012). A key advantage of this system is that the use of an open loop circuit cancels the need for cooling equipment, a disadvantage is that it is likely that filters have to be cleaned more frequently.

M. Schaap. Seawater Driven Piling Hammer, IHC Hydrohammer. In Proceedings of the Dutch Fluid Power Conference in Ede, September 2012. (reference added)

6. [RC1] "As it is proposed to use only one turbine and generator, reliability of these becomes a critical issue. Has the author any thought on this that he would like to share with the readers?"

[AC] Indeed, by using only one or a few turbines and generators, the reliability of these components become an important aspect. Modern hydro-turbines have been developed with typical capacities of 500 MW operating for decades with enough operational and maintenance experience gained from conventional hydro-power plants. On the other hand using hydro turbines in combination with renewable energy sources such as offshore wind energy has not been explored. The concept itself is still in predevelopment phase and therefore there is a lack of real data supporting the reliability. It is also expected that by having the whole electrical generation equipment in one offshore central platform instead of having it in a constraint space hundred meters above sea level, would have a positive impact regarding O&M costs.

7. [RC1] "Eqs. 6-7: The notation is slightly confusing. I assume that V(e) is a function depending on the variable e, later shown in Eq. 8. However, also other terms in Eqs. 6-7 are functions that depend on parameters. To be consistent, I suggest that you simply use V in Eqs. 6-7 and clarify V(e) = eVp;max in Eq. 8."

[AC] Equations have been modified accordingly.

8. [RC1] "Section 2.1.3: The pitch actuator model is based on a proportional regulator. Why not also a derivative or integrator component? Why is the pitch actuator model needed?"
[AC] The pitch actuator model is needed to account for any blade-pitch actuator dynamic effects. This means the slow or fast response of the pitching mechanism to the control command signal. The derivative or integrator components are considered to be included in the pitch control which is in series with the pitch actuator, see Section 3.3.

9. [RC1] "Section 2.3: The nozzle length Lnz should be indicated in Figure 4 as well." [AC] Figure 4 is now modified including Lnz

10. [RC1] "Section 2.4: What is the value of the vena contracta coefficient used here?" [AC] A value of Cv=0.99 was used according to (Thake, 2000). Please note that the vena contracta phenomenon does not influence the nozzle efficiency.

Thake, J. The Micro-hydro Pelton Turbine Manual. Practical Action Publishing, 2000. (reference added)

11. [RC1] "Section 3.1: 'A low pass filter on the pressure measured is employed' What are the filter characteristics?"

[AC] A first order low pass filter was used with the following transfer function form: LPF(s)= 1/(1+s/wc) where the cut-off frequency wc was set at 32pi [rad s-1]. This description is now included in the manuscript. Table 1 was added with the parameters of the augmented controller.

The author hopes that the modifications to the manuscript and replies to the reviewers satisfy your requests.

---

## Author Comment (AC3) · 24 Apr 2017

Thank you to the reviewer for the comments on the paper. The specific replies to the reviewer comments [RC2] are addressed by the author [AC] in the following lines:

1. [RC2] "Introduction, Page 2: the paper should present a more detailed overview of work carried out in this area so far and what new work is being presented in this study."

[AC] The author agrees, similar comment was done by RC1. The following paragraph has been added: The modelling and analysis of a single turbine with hydraulic technology has been previously presented for variable-speed control strategies. Simulations of an individual turbine with an oil based hydrostatic transmission have been presented in (Jarquin Laguna et al., 2014). The results showed good dynamic behaviour for turbulent wind conditions where reduced fluctuations of the drivetrain torque and power

are obtained despite the reduced energy capture. The integration of a single turbine with a Pelton runner using water hydraulics was introduced in (Jarquin Laguna, 2015), where a passive variable speed strategy was proposed. However, the addition and simulation of more turbines to the hydraulic network was not included. In an effort to assess the trade-offs implied by the proposed hydraulic concept, this paper extends the time-domain simulations to evaluate the performance and operational parameters of five turbines coupled to a common hydraulic network for a hypothetical wind farm with centralized electricity generation. In the first part of this work, an overview of the wind farm model is presented together with the control strategy of the hydraulic components; the second part describes a case example where the results are compared with those of a typical wind farm based on conventional wind turbine generator technology.

2. [RC2] "Page 2, line 8: amend sentence to end as follows: 'where the results are compared with those of a typical wind farm based on conventional wind turbine generator technology'"

[AC] Sentence has been amended.

3. [RC2] "Page 3: Equation (3) is missing. The equation number (3) is being indicated.

[AC] The equation indicating the aerodynamic power is now included.

4 to 9. [RC2] Specific comments to the text.

[AC] All the corrections to the text have been incorporated in the revised version.

10. [RC2] "Page 4, Eqt (12): Section 2.1.4 should include a brief explanation of how eqt (2) is used in conjunction with eqts (1,2) to determine the rotor torque."

[AC] . The following explanation has been added: The thrust force is calculated through Eq. (2) using the tip speed ratio from Eq. (4) and the rotor speed obtained from the solution of Eq. (5).

11. [RC2] "Page 5, section 2.2, line 17: it should be clarified in the text that linearity

only holds for laminar flows. For turbulent flow, the non-linear equations have to be applied."

[AC] . The following lines and references have been added in the revised document:

The linear models are only given for laminar flow, for steady flow the criteria for occurrence of turbulence is simply given by the Reynolds number; however, for unsteady flow neither the criteria used to predict flow instability, nor the manner in which it occurs is well understood. In the case of an oscillating flow component which is superimposed on a mean turbulent flow, the laminar flow solutions might be still applicable over a limited turbulent flow range. Both physical and empirical-based corrections to the shear stress model have been proposed for turbulent pipe transients (Vardy et al., 1993; Vardy and Brown, 1995). The correct modelling of turbulence in transient flows is an ongoing research topic; it is not addressed in this work.

Vardy, A. and Brown, J.: Transient, Turbulent, Smooth Pipe Friction, J. Hydraul. Res., vol. 33, p.435–456, 1995.

Vardy, A., Hwang, K., and Brown, J.: A Weighting Model of Transient Turbulent Pipe Friction, J. Hydraul. Res., vol. 31, p.533–548, 1993.

12. [RC2] "Page 7, line 16: a more elaborate explanation is required about the fundamental physics governing Pelton wheel operation: If the rpm is kept fixed, then the jet velocity and hence the press drop across the nozzle should be also fixed. k should also be fixed at the optimal value of 0.5 for optimal efficiency. An explanation of how this condition is applied in the numerical solution is necessary."

[AC] . A new paragraph and figure 6 have been added to clarify the Pelton operation: The theoretical Pelton efficiency is shown in Fig. 6 for different friction factors and constant bucket angle. Optimal efficiency is obtained when the water jet velocity is twice the tangential velocity of the runner at PCD. If the Pelton runner speed is kept constant, then the jet velocity and hence the pressure drop across the nozzle should

be also kept constant in order to operate at maximum efficiency. A Pelton turbine operating with a constant rotational speed considerably simplifies the integration with the electrical grid. The constant rotational speed is realized by using a grid-connected synchronous generator, similar to most large scale hydroelectric generation plants.

13. [RC2] "Page 9, first line: 'so-called' "

[AC] Correction is made in revised version.

14. [RC2] "Page 9, line 9: this is linked to comment 12 above. Explain in the text why you have a constant pressure supply. To what extent is the control system able to maintain a constant pressure when intermittent wind conditions cause the water flowrate to change abruptly?"

[AC] The following explanation is now included in section 3.3 Pump controller: A constant pressure in the hydraulic network is desired, not only to keep the Pelton turbine operating at maximum efficiency as described in section 2.4 , but to be able to connect the water pumps from the individual turbines to the hydraulic network. In addition, maintaining a constant pressure supply is beneficial in minimizing fatigue damage to the hydraulic system components

15. [RC2] "Page 9, section 3.2, first line: it is worth mentioning that maintaining a constant pressure supply is beneficial in minimising fatigue damage to the hydraulic system components."

[AC] Added, see previous comment.

16. [RC2] "Page 19, first line. Explain the difference between the control systems of Buhagiar et al and that being proposal here."

[AC] The following clarification is now included in section 3.2 Spear valve controller: Another option is to implement a constant pressure control as proposed in (Buhagiar at al, 2016), where a feedback controller is used in combination with feed-forward compensation.

17. [RC2] "Page 11, line 9: Include a table with the derive values for the different gains."

[AC] Table 1 with Controller parameters of the spear valve augmented controller is now added.

18. [RC2] "Page 12, line 15: wouldn't a compressed air or weighted accumulator help solve the problem of increased activity of the pitch controller?"

[AC] This could be possible if a pneumatic or hydraulic actuator is used for the pitching system, however in this work the pitch actuator and controller is maintained from the NREL reference turbine without modifications to allow an easier comparison of the proposed hydraulic concept.

19. [RC2] "Page 13, Figure 10: if the hydraulic turbine only includes an open-loop system with the pump housed at the nacelle, then a separate boost pump is required to be able to supply the sea water up the hub height. Has this been factored in the analysis?"

[AC] This is correct, the following lines have been added for clarification: For the hydraulic turbines, a separate boost pump is required to supply the water to the pump located at the nacelle. Together with the filters and cooling system these components comprise the auxiliary equipment which is not included in the analysis. The same consideration is made in the conventional wind turbine technology where the lubrication, filtering and cooling power required by the gearbox and generator is not included in the analysis.

20. [RC2] "Page 18, line 1: add a full stop – 'conventional technology. For the presented..' "

[AC] Sentence has been corrected.

21. [RC2] "Page 18, line 3: quote here the percentage efficiency of the pump and that of the hydraulic network. "

[AC] Both efficiencies are included in the conversion from point A to point B. Making distinction of the separate efficiencies is in the author opinion more confusing and not consistent with the rest of the presented results.

22. [RC2] "Page 18: Conclusions - comment on any opportunities for costs reduction offered by the new concept".

[AC] Opportunities for cost reduction are now included in the introduction with references. The conclusions also include further work.

"Hydraulic systems have already shown their effectiveness when used for demanding applications where performance, durability and reliability are critical aspects. In particular, the efficient and easy generation of linear movements, together with their good dynamic performance give hydraulic drives a clear advantage over mechanical or electrical solutions. Furthermore, hydraulic drives have the potential to facilitate the integration with energy storage devices such as hydraulic accumulators which are important to smooth the energy output from wind energy applications (Innes-Wimsatt and Loth, 2014). In any industry where robust machinery is required to handle large torques, hydraulic drive systems are a common choice. They have a long and successful track record of service in, for example, mobile, industrial, aircraft and offshore applications (Cundiff, 2001; Albers, 2010). Therefore, it is evident that the use of hydraulic technology is recognized as an attractive alternative solution for power conversion wind turbines (Salter, 1984). For the proposed concept, using high pressure makes it possible to reduce the top mass of the individual rotor-nacelle assemblies. For this reason, a high potential exists to reduce the amount of structural steel needed in the support structures as well; for a 5 MW turbine in 30 m water depth, 1.9 ton of structural steel of the monopile can be saved for every ton of top mass reduction (Segeren and Diepeveen, 2014). Using high pressures makes the use of fluid power an attractive means to transmit the captured energy from the rotor-nacelle assemblies to a central platform."

The author hopes that the modifications to the manuscript and replies to the reviewers satisfy your requests.